# Narratives of Success and Failure in Ressentiment: Assuming Victimhood and Transmuting Frustration among Young Korean Men

Tereza Capelos [1,*], Ellen Nield [2] and Mikko Salmela [3,4]

1   Political Science and International Studies, University of Birmingham, Birmingham B15 2TT, UK
2   Liberal Arts and Natural Sciences, University of Birmingham, Birmingham B15 2TT, UK
3   Practical Philosophy, University of Helsinki, 00014 Helsinki, Finland
4   Center for Subjectivity Research, University of Copenhagen, DK-2300 Copenhagen, Denmark
*   Correspondence: t.capelos@bham.ac.uk

**Abstract:** In this article, we examine toxic masculinity, anti-feminist, anti-globalisation, and anti-military conscription positions in the narratives of what constitutes success and failure among young South Korean men during the COVID-19 pandemic. Misogynistic accounts attributed to the globalised effects of neoliberalism and its evolution through South Korean meritocratic competition, compounded by the social isolation of the pandemic, remain a puzzle psychologically, despite their toxic emotionality. We use the analytical framework of *ressentiment* to consolidate references to moral victimhood, indignation, a sense of destiny, powerlessness, and transvaluation, as components of a single emotional mechanism responsible for misogynistic accounts. In an empirical plausibility probe, we analyse qualitative surveys with young South Korean men and examine the content of the far-right social sharing site Ilbe (일베) which hosts conversations of young men about success and self-improvement. Our findings show envy, shame, and inefficacious anger transvaluated into to moral victimhood, misogynistic hatred, vindictiveness against women and feminists, and anti-globalisation stances. We discuss how the content of these narratives of success and failure in *ressentiment* relates to the electoral win of the right-wing People Power party in March 2022 which capitalised on anti-feminist grievances. We also consider the socio-political consequences of *ressentiment* narratives in the highly gendered and polarised South Korean society and expand the study of *ressentiment* outside the context of Western democracies where it has been most extensively elaborated.

**Keywords:** *ressentiment*; South Korea; COVID-19; misogyny; Incel; victimhood; powerlessness; anger

## 1. Introduction

In this article, we examine understandings of success and failure among young men in South Korea during the COVID-19 pandemic, and we use the analytical framework of *ressentiment* to elaborate on the highly gendered and grievance-laden content of these narratives. Deepening age and gender divides became particularly relevant during the March 2022 South Korean Presidential Election as the right-wing opposition People Power party led by Yoon Seok-Yeol defeated the incumbent left-wing Democratic party led by Lee Jae-Myeong. Electoral results showed significant polarization among South Korean men and women in the 'under thirty' age group where Yoon Seok-Yeol was supported by 58.7% of men compared to 33.8% of women (Yoon 2022). Recent studies noted a culture war was fought online on young people's websites such as Womad and Ilbe, where 'trolls' 're-inforced an us-and-them rhetoric, divided along gender lines: man versus woman' (Koo 2019, p. 835). While some young people shared views on politics and society often through humour, irony, and hyperbole, similar to 4Chan or Reddit, others engaged in narratives of success and failure centred around toxic masculinity, sharing anti-feminist, anti-globalisation, and anti-military conscription positions (Um 2016; Yun 2013). Studies of

young people inhabiting exclusionary online communities also showed that women on Megalia and Womad imitated the hyperbolic, ironic, and shocking rhetoric, originally characteristic of male online communities like DC Inside and Ilbe (Jeong and Lee 2018; Kim 2021). We are interested in understanding how misogynistic performances of gender politics online and offline are expressed by young men in South Korea through narratives of success and failure.

Narratives of how people make sense of their lives 'are often stimulated or catalysed at points of crisis and tension' (McAuley and Nesbitt-Larking 2022), and our study took place during the global pandemic context which had profound implications for the daily lives of millions. Mobility restrictions placed by requirements of social isolation and quarantine posed physical, economic, and political stresses (Haleem et al. 2020). In South Korea, surveys highlighted the association between depression and continuous restrictions of daily life impacting physical activity, sleep, nutrition, and stress management (Cho et al. 2022). Studies also noted that the pandemic was linked to the proliferation of polarising narratives globally. The psychological toll of social alienation experienced by many has given rise to mental health problems such as depression and stress, domestic violence particularly against women and minorities, and online radicalisation by far-right-wing, white supremacist, and (violent) extremist movements, harnessing old and new grievances and frustrations (Ariza 2020; Fitzpatrick et al. 2020; Gunraj and Howard 2020). In this context, narratives of hostility, distrust, and stigmatisation, and deepened social divides, have been exploited by political parties and leaders for electoral gain (Kinnvall and Singh 2022). Furthermore, the impact of the pandemic on young people has been profound. Mobility restrictions impacted social interactions and extracurricular activities, compromised young people's sense of autonomy and freedom, promoted a sense of forced seclusion, and moved social life online to digital spaces, for some almost exclusively (Kelley 2020; Morton et al. 2021). During the pandemic, 'angry young men' spent more time in spaces such as gaming platforms, chat rooms, blogs, and forums of the 'manosphere' (online spaces dedicated to masculinity and men's issues, often toxic and primarily misogynistic), a practice which exacerbated their social isolation and alienation (Basu 2020; Russell and Bell 2020). It is in this context that we seek to understand how misogyny permeates narratives of success and failure among young men in South Korea. Extant studies offer two key structural factors as its explanations: globalisation and military conscription.

Several scholars argue that neoliberalism is destabilising the identity of young men (Baer 2016; Kim 2001, 2018, 2015). South Korea has rapidly developed under the shadow of American neoliberalism, becoming a hyper-competitive society, where collectivism and neoliberal capitalism combine (Kuznets 1981). The change in political ideology from traditional collectivism to neoliberalism places responsibility for success and failure solely on individuals (Mounk 2017; Sandel 2020) and influences how young people in a highly competitive society like South Korea view success, failure, and identity (Kim 2009; Ratner 1997; Smith 1996).

Gender relations in South Korea centre around discussions of racial and sexual inferiority, following from the gendered divide of military conscription (Choo 2020). Male citizens between the ages of 18 and 35 must perform compulsory military service for 18 months to 2 years, while their female counterparts are exempt. Conscription has become the main catalyst for gender conflict in South Korea, especially among young people (Choo 2020; The Economist 2021; Kwon 2000, 2021). In this context, some young men express anger against women who are perceived to have stolen masculine supremacy, originally deemed a pre-existing right (Song 2014). These structural explanations, albeit important, do not address the psychological content of these often-misogynistic narratives of success and failure and the emotional experiences attached to them.

We bring together insights from studies in political psychology, feminism, sociology, philosophy of emotions, narrative analysis, and Korean studies, to understand the affective undercurrent of what constitutes success and failure among young South Korean men. We approach anti-feminist, anti-globalisation, and anti-military conscription accounts as

interwoven narratives bundled together by *ressentiment* affect. *Ressentiment* denotes a long-lasting emotional experience which contains victimhood, envy, a deep sense of destiny, injustice, and powerlessness transmuted into other-targeting negative emotions like hatred, resentment, and vindictiveness (Salmela and Capelos 2021; Demertzis 2020; Salmela and von Scheve 2017, 2018). We use *ressentiment* as the analytical framework to understand expressed insecure and grievance-laden misogyny against women as well as anti-globalisation and anti-feminism narratives. *Ressentiment* has been studied in the context of grievance politics and far-right movements, predominantly in the West, focusing on the 'left behind', the economically or socially disillusioned, the white, angry, and fragile males (Brown 2019; Capelos and Demertzis 2022; Kay 2021; Kimmel 2017; Salmela and Capelos 2021), and the 'losers of globalisation' (Akkerman et al. 2013; Mishra 2017). Here, we employ it beyond the Western context, in the cultural and political environment of South Korea. Furthermore, while insecurity has often been cited as a reason why some individuals are misogynistic (Banet-Weiser 2018; Ging 2017; Levit 2020), our account approaches insecurity, as well as grievances and frustrations, as triggers of *ressentiment*, which in turn generates misogyny and other anti-social outcomes, adding analytical clarity and highlighting the rich emotional layers of this process.

### 1.1. Narratives of Success and Failure among Young South Korean Men

To understand the content of accounts of success and failure among young South Korean men, we take stock of the distinct social, cultural, and historical elements that shape these narratives. South Korea's youth have had a unique cultural upbringing with a fast turnaround to modernisation, impacting their self-belief in their own skills and competence. Walkerdine notes intensified privatisation, individuation, and globalisation lead to individuals feeling they must 'produce [themselves] as having the skills and qualities necessary to succeed' (Walkerdine 2003, p. 240). In a hyper-competitive market, creative, global youth have been critical to reforming the economy (Song 2003), and this meritocratic-competitive model in South Korea has been linked to collectivism, especially in education and employment (Kim 2009; Smith 1996). Extant studies suggest this model occurs under the influence of neoliberalism from the USA and has led to a change in the cultural success index for young South Koreans, defined by personal ability, responsibility, style, and consumption (Abelmann et al. 2009). Taken together, these factors of neoliberal modernisation and traditional Eastern collectivism create in South Korea a unique melting pot of individual responsibility to achieve and collective burden to succeed (Hairong 2003; Anagnost 2004; Mounk 2017).

There is a distinct difference in the way success is constructed between young South Koreans and their parents and grandparents, due to this changed political philosophy brought about by rapid modernisation and neoliberal influences. The Democratisation Generation, instrumental for the democratic transformation in the 1980s through student activism and protests, was succeeded by Generation X, the country's first post-democratisation generation. They were less active champions of democratisation and have been serving as a bridge between traditional collectivist nationalism and hierarchical organisational culture and the younger New Generation's individualism (Kang 2020; S. J. Lee 2021). The individualised sense of personhood in a hyper-capitalistic and fast-expanding market (Comaroff and Comaroff 2000) hides the structural forces that can constrain futures (Abelmann et al. 2009, pp. 232–33). South Koreans currently in their 20s and 30s are endowed with a skewed sense of meritocracy and the belief that society has fewer social ills due to the impartiality of competition, brought about by catch-up modernisation (S. J. Lee 2021). In this context, constructions of masculinity are linked to neoliberalism and modernisation considerations (Choo 2020; Kwon 2013; Kim 2001). The neoliberal model assumes that men and women are equal, and therefore there is no discrimination. Studies show, however, that changes to correct any imbalances through the redistribution of income or extending welfare create contention and feelings of victimisation among men (Choo 2020; Hubbard 2004).

Along these lines, as military conscription only affects young men, it is felt as a hindrance in achieving higher education and entering the job market and generates frustrations (Kim and Finch 2022). Compulsory military conscription is a unique aspect to life for young South Korean men, not shared by women, the experience of which contributes to the nursing of grievances about this obstacle which delays or inhibits self-actualisation (Choi and Kim 2016). As studies note, this male practice of military conscription is tied to anger, violence, masculinity, and citizenship (Agostino 1998; Choi 1997; Kwon 2000; Moon 2005). Sociological and feminist literatures on conscription examine its impact on perceptions and expressions of masculinity, focusing on how aggressive behaviour, violence and anger are instilled in young men within a patriotic and gendered setting (Choi 1997; Sherman 2007). Studies also show aggression is aimed towards women due to gendered isolation and the 'effeminate' becoming something 'other' that must be banished (Agostino 1998; Moon 2005; Theweleit 1989). This is all discussed within the context of citizenship and patriotic sacrifice (Enloe 2000; Kwon 2000; Moon 2002; Yuval-Davis 1997). Mandatory conscription in South Korea is a national service, perceived as a sacrifice. As such, aggression, banishment of femininity, and patriotism are all compounded and instilled at a foundational time of life for young men. Nearly all young men in South Korea go through this coming-of-age ritual, which carries with it grievance, frustration, and a heightened sense of victimhood.

A parallel, perhaps not unrelated source of frustration is sex. In recent years, scholars have noted an increased interest in Incel (involuntary celibate) cultures where young men become angry and resentful when denied sex by women and create communities to vent these frustrations in 'networked misogyny' (Banet-Weiser 2018). The causes for the manifestation of Incels ranges from 'scopic capitalism' and the value of sexual capital (Illouz 2019), to narcissistic regressions of white supremacy and anti-feminist backlash (Hoffman et al. 2020), and to outcomes of a Tinder-fied dating culture, where men have a lower sexual market value (Bloodworth 2020). This complex mixture of factors that lead to men feeling frustrated about the denial, perceived or actual, of sex, is at its core associated with the loss of a pre-existing sense of entitlement, not necessarily about sex. However, inferiority narratives intertwined with sex are dominant in the male-identified Incelosphere, attaching anger onto a 'misandrist' society, perceived to be the ultimate cause of humiliation and despair. Blame is apportioned to an external cause, as hatred and anger are redirected against women and feminists (Bratich and Banet-Weiser 2019; Kay 2021, p. 38). Feelings of victimisation follow on from the denial of a sense of entitlement (Illouz 2019), and we expect they become a frequent consideration in what is perceived as failure.

*1.2. Ressentiment as the Framework of Understanding Misogyny in Accounts of Success and Failure*

Anti-feminism, anti-globalisation, and anti-military conscription positions among young South Korean men are often expressed as negative affects towards women and feminists. These culturally reinforced positions could be evidenced in accounts of success and failure online and offline. To understand the deeper psychological properties of misogyny, we employ *ressentiment* as the analytical framework which can highlight how insecurities, grievances, and frustrations of inadequacy can be transmuted into other-targeting negative emotions, in this case against women and feminists, and transvaluated to a morally righteous sacrifice. *Ressentiment* is not a new term. It denotes a psychological experience introduced by Nietzsche ([1884] 1961) and studied further by Scheler (1961), and has been described as a 'cluster' of emotions (Capelos and Katsanidou 2018; Demertzis 2004, 2006, 2020; TenHouten 2018), a compensatory defence mechanism (Salmela and Capelos 2021) or a dispositional state of being (Rodax et al. 2021).

Distinct from resentment which denotes anger, *ressentiment* is triggered by repeated failures or losses that lead to frustration (Hoggett 2018; Demertzis 2020; Salmela and Capelos 2021). According to Salmela and Capelos, at its core, *ressentiment* is a double transvaluation emotional mechanism, often likened to Aesop's fable of the Fox and the Sour Grapes. Following repeated failure to acquire what is desired, the self is re-evaluated from origi-

nally being a loser, to being a pious victim, and the object that was originally coveted and highly valued is re-evaluated to be an object that is no longer sought after. *Ressentiment* distorts reality, but its distortion has a compensatory, defensive function: when an individual is unable to cope with the failure or loss, the negative feelings associated with this repeated deprivation such as impotence or inferiority are bypassed by adopting an alternative, less painful account of events. *Ressentiment* at its early stage can serve as a temporary resolution to cope with emotional pain, but it can also develop into a chronic position. At its advanced stage, the individual rejects the inferior 'old self' and its 'old values' as all-bad and adopts a self-righteous victim position, perceived as 'all-good'). The new beliefs about the self and what it values can be reinforced, cemented, and sustained through social interactions with others (Salmela and Capelos 2021, pp. 197–99).

In the South Korean context, cultural and political factors such as the rapid modernisation under the conflicting ideologies of American neoliberalism and traditional collectivism seem to contribute to young men feeling victimised by feminism, enhancing isolated masculinity through military conscription, and generating tensions around sex and relationships in a globalised society. Without seeking to examine how *ressentiment* is constructed, we employ it as an analytical framework to make sense of the emotional experiences of young men contemplating what is success and their capacity to cope with failure, loss, and frustration. We expect that if *ressentiment* is at work, we would see strong evidence of powerlessness, blended with inefficacious anger and victimhood, as well as evidence of transvaluation, pointing to an experience that becomes elevated from weak and self-reproaching, into morally righteous and other-blaming.

## 2. Materials and Methods

Our empirical investigation examines whether observed patterns in young men's narratives of success and failure are consistent with *ressentiment* using a plausibility probe. A plausibility probe is a stage of empirical inquiry preliminary to testing, which examines the 'plausibility' of a theory. Empirical plausibility probes adopt suggestive tests, do not require large representative samples, and establish whether a theoretical construct is worth considering, without providing exact estimates of probability (Eckstein 1992).

Our empirical plausibility probe involves two complementary data sources: a qualitative survey of young Korean men, and a thematic analysis of conversations hosted on the far-right South Korean social network website Ilbe. In the qualitative survey, we focus on how young South Korean men discuss success, aspiration, and identity. Through the analysis of Ilbe content, we examine representations of what is personally and culturally coveted in the context of online discussions about feminism.

### 2.1. Qualitative Survey Sampling and Design

We recruited participants for the qualitative survey from Yonsei University and Konkuk University student chats on Kakao Talk and on the social media site Reddit under the subreddit forum called r/Korea. Our sampling frame required South Korean men under forty. Of the nineteen respondents that completed the survey, twelve fit this profile. While this sample is not large, it allowed to explore whether *ressentiment* as an analytical framework is helpful for understanding the anti-globalisation, anti-feminism, and anti-military conscription accounts discussed in the literature. Our topic was potentially sensitive, and we opted for an open-ended qualitative survey instead of interviews, with the following considerations in mind: a self-administered questionnaire allows for higher validity in responses and more flexibility in information gathering by promoting a sense of individuality, visual anonymity, and the comfort of being able to complete the survey at one's own time and read and write in one's own native language. It also reduces the risk of participants' self-consciousness during interviews influencing responses (social desirability), and the risk of the characteristics of the interviewer (age, gender, nationality, race, interaction style, language accent, foreign appearance) influencing the outcome of interviews (interviewer bias) (Albudaiwi 2017; Rossi et al. 1983; Sicmiarycki 1979). About half of the respondents

(referred to here as R1, R2, etc.) completed the questionnaire in English and the other half in Korean. A native Korean speaker checked the translation from Korean to English (see Appendix A for the survey questionnaire).

The questionnaire contained twelve open-ended questions related to individual and group identity in the context of South Korean society and politics[1], as well as views on success, failure, achievements, challenges, and aspirations[2]. Questions were worded with a personal marker, such as 'how would *you* define fairness?', inviting respondents to highlight factors salient to them personally rather than generic understandings of fairness. We did not ask questions about the pandemic and its impacts as we did not want to prime participants to think of success and failure in a particular context. We employed thematic coding of the responses which yielded nine categories: challenges one faces, unique factors of South Korea, the politics of South Korea, relating to others, comparing oneself to others, defining fairness, defining success, defining failure, and important issues. The pandemic was mentioned by R4 in their answer on recent challenges they faced and what they have done to overcome them. This respondent answered 'COVID has been the final nail in the coffin for my CS education. Online classes have been harsh on me as an adult ADHD sufferer. I've temporarily given up my dream of being a developer and elected to grab IT job certs.' This statement shows how the pandemic impacted this individual's ability (or perceived ability) to achieve a goal. We think most respondents faced similar experiences and challenges, although they did not choose to report the pandemic as one of their identified challenges. While we cannot empirically test this hypothesis, we consider it likely given the impact the pandemic had globally, and in South Korea in particular, which imposed strict limitations on social gatherings, electronic log system requirements, business hour restrictions, and prohibitions on food consumption in public places, affecting social activities and quality of life overall (Hyun et al. 2021). Taking these factors into consideration, it is plausible that the pandemic was a significant factor contributing to the frustrations and grievances that raised young people's *ressentiment*.

The questionnaire also contained a six-item ten-point Likert scale (strongly disagree to strongly agree) of *ressentiment*, used in Capelos and Demertzis (2022). The scale items measured indignation, victimhood, powerlessness, transvaluation, injustice, and destiny: 'People enjoy better standard of living with less effort' (indignation); 'I feel that many people take advantage of my kindness' (victimhood); 'I feel I do not have people's respect' (powerlessness); 'Many seem important, but they should not get such attention' (transvaluation); 'Everything goes wrong in my life, why me?' (injustice); 'My hopes and dreams will never come true' (destiny).

### 2.2. Content Analysis of Ilbe Cases (일베)

For the content analysis, we collected data from the far-right South Korean community-led blog website Ilbe (일베) which is known for its misogynistic content (Um 2016; Yun 2013). In Ilbe, individuals share memes and funny anecdotes that reflect their views on politics and society. Ilbe is a male online space and bans female users from joining (H. Lee 2021). It is predominantly used by young South Korean men in their twenties and thirties, with 35% self-reporting being twenty-one to twenty-five years old in a 2013 poll (Pearson 2013). Our aim was to examine the messages of politically and socially marginalised users who anonymously engage in online social sharing with peers, focusing on feminism-related content. We used the prefix 'femi' ('페미') and collected the links and titles of a randomly chosen date in the year: 20 January 2022, focusing on the first 200 results. We had 156 useable posts of which we randomly selected 50 for which we extracted all text and images[3]. All content was translated back to English and coded, and cases are referred to here as C1, C2, etc. Although we did not search the platform for pandemic-related content, COVID-19 was mentioned in a few cases. C12 berates the government for not allowing an exception for the quarantine pass for pregnant women (the quarantine pass was proof of either vaccination or recent negative test, needed to use public spaces such as gyms, karaoke houses, and cinemas from December 2021 to February 2022, according to APNEWS.com

(APNews 2022)). This statement implies that pregnant women who got the vaccine would have given birth to children with deformities, and it calls for protests and an overwhelming regime change to end mask wearing and social distancing in the 'corona scam'. C19 states that the government is a 'left dictatorship' because 'judges oppose without reason quarantine pass exemptions'. C22 noted that the current politicians are populist cult leaders and an example to show this is the 'quarantine pass for students' (implying that the quarantine pass is a left-wing populist agenda and forcing it on students victimises those who are healthy). Finally, C32 stated that Korea is doomed no matter who they chose (in the March 2022 election) and that a reason not to vote for one candidate is that they are a 'vaccine defender'. Similar to the survey data, these unsolicited references to the pandemic in feminism-related content suggest that the pandemic and its social impacts was internalised by young South Korean people, and contributed to emotionally loaded, heavily politicised, gendered and conspiracy-laden narratives in the 'manosphere' (see also Wojnicka (2022) on hegemonic masculinities and the pandemic).

To code for *ressentiment*, we adapted the Capelos and Demertzis (2022) 6-item scale. For each case, we coded six discrete variables with value of 1 where there were explicit mentions or implicit accounts referring to indignation, victimhood, powerlessness, transvaluation, injustice, and destiny, and with value 0 where these concepts were explicitly or implicitly absent. We also coded for implicit content available through images, metaphor, or satire (see Appendix B for the variables coded in the content analysis, and Appendix C for coding examples of the first three cases).

To improve the stability, reproducibility, and accuracy of the coding (Krippendorff 1980, pp. 130–34), the coder was familiarised with the data, the literature, and the variable operationalisation and coded the material in a few days to avoid influences from external factors. We also took care to protect the psychological integrity of the coder. The content on the far-right site Ilbe (일베) was often hateful, misogynistic, and involved depictions of violence and sexual assault. It therefore required taking regular breaks and emotional distance from this material.

## 3. Results

### 3.1. The Content of Success and Failure: Understanding Frustrations and Aggressions

The survey contained questions about the essence of success and failure, challenges of achieving one's goals, and life in Korea. Respondents (R) made references primarily to happiness, respect, and social life, finance, occupation, and achieving their goals. Discussing financial success, R7 noted 'I want a large family and to be able to provide for it just as my parents have for my siblings and I'. R8 said 'I want to receive a real monthly salary of around two million won [approximately $1400], living in a house of a suitable size [. . . ] and enjoy small shopping and entertainment (around $200 a month)'. Objective goals like securing a house, spouse, job, or money were also markers of success across several respondents (R1, R2, R4, R6, R8, and R12).

Discussions of success raised anxieties about meritocracy. The question on unique factors of South Korea contained mentions of achievement: '[Korean's] tend to be competitive' (R1), 'overly obsessed with growth' (R3), 'sustained belief in capitalist mobility' (R4), 'desire for achievement' (R6). Hyper-competitiveness and a constant strive for upward mobility means there is always a constant goal ahead, and there is always something to be desired.

While neoliberalism masks structural inequalities (Y. Lee 2021), the belief of attainment that is characteristic of the South Korean meritocratic-competitive model exacerbates this masking even further. Believing an ambition or aspiration is achievable through self-improvement means that when it is denied, there will be far greater feelings of impotence and failure. This is because the unattainability of a goal that is viewed as attainable through self-improvement could be seen as a failure of the self through a lack of said improvement. By disguising social inequalities that may hinder achievement, there is an increased likelihood of misidentifying injustices, as there is no clearly identifiable object onto which to

re-attach feelings of failure. Therefore, the neoliberal meritocratic-competitive model that drives South Korean youth could lead to more opportunity for failure, increased negative emotions when success is denied, and no identifiable object for those emotions to be directed towards.

The definitions of success and attainment among South Korean youth have evolved since previous studies (Abelmann et al. 2009) and are characterised by individual responsibility and material gain through constant upward movement. We expected that the combination of these neoliberal influences in this hyper-competitive society would generate conditions that trigger *ressentiment*.

### 3.2. Frustrations of Military Conscription

The survey mentioned military conscription only once, and this was in relation to recent challenges. R1 stated that 'due to mandatory military service in Korea, there was a risk of academic interruption'. They solved this problem by using an alternative system of service personnel by passing an exam, so instead of serving in the military they served in the police. This shows that military service, at least by some, is seen as a challenge to overcome, particularly in relation to academic attainment.

In the content analysis of Ilbe, military service was frequently mentioned as a barrier towards individual goals, redirecting frustrations towards women and feminism. Conscription was the main topic in four different cases (8%) and mentioned in nine (18%). Highlighting the link between feminism/feminists and military service, military conscription was discussed as a valiant hardship that must be endured, while women, feminists, and politicians make soldiers suffer. Case 4 references a conversational YouTube channel made by a Korean man who serves in the military and describes his service: 'I almost think that I am the lowest class in society [. . . ] I wake up every day at minus 20 degrees, 40 degrees below a sensible temperature and I freeze up and work for several hours at my post [. . . ] No matter how much service time, I still write a letter. If the social atmosphere had the least respect.' This case is referencing letters sent from students at an all-girls high school (Jinmyung school in Mokdong) to men completing military service. The school gave young female students the assignment to write supportive letters thanking conscripts for their service, but some students produced letters that were derogatory. In response, online anti-feminists advocated violence and sexual assault against the students[4].

An important question here is how frustrations about military conscription relate to success and failure. Conscription of course is not a failure as such but could relate to failure for these young men who experienced the mandatory service (from which women are exempted) as an obstacle to their success, defined in educational and material terms. This experienced injustice elicited a sense of victimhood, and as we would expect in *ressentiment*, the negative self-focused emotions about actual or anticipated failure in competition were redirected onto others. While we cannot definitively say why conscription (which involves state-forced isolation in a violent and male-dominated space) was not mentioned frequently in the qualitative survey but was featured frequently in Ilbe, we can provide a tentative explanation: online environments like Ilbe facilitate sharing narratives of victimhood (in this case about conscription) and redirect negative affects onto others. A likely explanation is that the strictures of the COVID-19 pandemic heightened the sense of victimhood about conscription among these young South Korean men, similarly to how the pandemic and its strictures contributed to emotionally loaded, heavily politicised, gendered and conspiracy-laden narratives in the 'manosphere', as we have observed above.

### 3.3. Narratives about Sex

The survey and Ilbe data link comparisons to others to sex, and through sex, to race and nationality. Rarely is sex mentioned in the context of 'romantic and sexual fulfilment' (Survey, R4). The Ible data reference sex as sexual capital, with frequent mentions of sexual appeal and attractiveness. In the 50 sampled cases which included the prefix 'femi', about 32% (16) referred to sex, rape, or sexual appeal. In this subset of cases, when discussing sex,

the emphasis on race and nationality was 8% higher than the sample overall (see Table A3, Appendix D). Men often expressed feeling victimised by women as they are 'so cheap to men in your country. Fucking bitches who acted like I'm nothing' (C14). The nationality aspect of identifying as South Korean men being treated poorly by women, combined with 'being overlooked as a sexual partner' through racial inferiority (C18), implies that this feeling of victimhood is not related to individual identities, but links to collective identities of being Asian and South Korean males. In this subset of cases focusing on sex, sexual appeal, or rape, blame was more heavily attributed to women compared to the sample overall, while feminists were equally blamed.

*3.4. Recognising Ressentiment*

Looking more closely at the content of frustrations expressed by young South Korean men in the Ilbe data, we identified references to powerlessness ('if the social atmosphere had the least respect', C4), morally charged victimhood (referring to the gruelling conditions of service that is demeaning), injustice ('I almost think I am the lowest class in society', C4), and destiny for sacrifice, particularly in reference to military service.

In the survey, the most frequent elements of *ressentiment* were indignation (other people enjoy a better standard of living with less effort), injustice (everything in life goes wrong, why me?) and destiny (my hopes and dreams will never come true). *Ressentiment* was evident also in the open-ended responses of participants. For example, when asked about the roles of others in society, R4 said it was to 'uphold a crumbing, exploitative structure that will leave their precious pensions in the dust' and R8 said 'the poor donate money to the rich and compete with each other to see who has contributed the most, and the rich play the role of multiplying their wealth with wealth'. These examples point to an individual sense of morally enhanced victimhood, where those who contribute to society are blind to the exploitation of the elite class, and they are the only ones who can see the injustice. These examples also contain a hint of superiority, which aligns with conceptualisations of *ressentiment* described in the literature.

In the Ilbe data, of the six markers of *ressentiment*, we counted on average 1.74 markers per case. In about a third of the cases (30%), we identified three or more markers of *ressentiment*, and 12% contained four or more markers. The most frequently occurring markers of *ressentiment* were powerlessness in 74% of cases, and victimhood in 32% of cases. This aligns with theoretical expectations of the significant role of these two markers for *ressentiment* (Salmela and Capelos 2021).

We also examined whether the negative feelings resulting from the perceived injustice or victimhood were projected against a particular target. In three survey respondents (R4, R17, R27), the focus for injustice/victimhood was the Mokdong/Jinmyung all-girls high school incident. Three respondents (R24, R29, R46) referred to a general lack of respect and ridicule from women, two respondents (R22, R23) referred to the salary of 2 million won a month for a conscript, and one respondent (R50) referred to military service as communism. Five respondents (50%) named feminists/feminism or women as they re-attached blame for their negative affects (R17, R24, R27, R29, R50), and five (50%) named a political individual or party (R4, R22, R23, R46, R50). R50 listed both. While we do not see a uniform source for the negative affects caused by military conscription or the treatment of soldiers, the vilification of women, feminists, and the political elite is clear. For young South Korean men who feel victimised by military service, there seem to be two routes for blame: feminists or politicians. The former connotes a militarised gender divide, and the latter indicates a neo-liberal mindset—both linked to feelings of victimhood.

Coding for the political orientation of the sources of victimhood and discontent, we noted that they span from the far-right to the far-left (Table A4, Appendix D), linking victimhood to discontent expressed towards the entirety of the system, rather than a particular ideology. It appears that the young South Korean men in our study, living within a neoliberal model which still retains tensions from the remnants of collectivist ideology,

expressed victimisation, noting the barriers they themselves face, but not mentioning the barriers that others face.

In the subset of cases related to sex, there were more mentions of *ressentiment* (9% increase, Table A3, Appendix D) compared no non-sex related cases, through references to denied entitlement and feelings of racial inferiority. Feelings of impotence were projected onto women, creating a sense of victimhood. C18 describes women as shallow for being 'no longer satisfied with their relationships with men of other races' once having sex with a Black man due the size of their penis. Here we see an example of feeling victimised by women due to a perceived notion that women devalue men based on race and penis size. Women, more so than feminists, are the outgroup and source of blame, suggesting it is the fault of all women for these men's feelings of victimhood.

Another aspect of *ressentiment* is the devaluation of what is coveted to being unwanted and unworthy of attention. The analysis of Ilbe material highlights how women being sexually attractive or being used as sexual objects was approached as a pre-existing entitlement for these young men. Five cases explicitly state that feminists are not sexually arousing or attractive. Some are simply devaluing feminists as not being attractive: 'Femoid features. Not sexually arousing' (C13); 'Ugly women deserve feminism' (C15); 'Before and after dying with Femi. So damn ugly' (C31). C43 expresses sadness and anger that a K-Pop girl group is not scantily clad as they have 'succumbed to their [feminists] power' by putting on suits. The narrative throughout is one of disappointment or disapproval that women are not attractive to these men.

## 4. Discussion

Our analysis of narratives of success and failure among the young men that answered our survey showed that success in South Korea is frequently associated with financial gains. Abelmann et al. (2009) describe the intersection of collectivism and neoliberalism as creating young South Koreans who 'aspire so eagerly to an individualised project of human development' (p. 243). The last decade has seen an evolution in the aspirations of young South Koreans, where some still hold an individualised goal that helps the collective, while others have a personal goal that is defined by consumption and material gain. Our findings fit this observation.

Contemplating success among participants of our survey also brought about anxieties. Extant studies identify the product of a constant need for a larger goal as another effect of neoliberalism in the meritocratic-competitive model. Neoliberal influences from the USA have changed the success index towards personal responsibility (Abelmann et al. 2009). The weight of personal responsibility combined with the collectivist basis for the meritocratic-competitive model that South Korea currently works under (Kim 2009) creates a burden of success that is individualised and never satisfied. Our data also point to this direction, and we think this is a finding that warrants further investigation. It would be valuable to examine whether the collectivist basis of the South Korean meritocratic-competitive individualist model aggravates or alleviates this individualism, and how young men express collectivism in their personal and political lives.

Turning to the Ilbe material, discussion of frustrations frequently involved militarised masculinity. The hardships of military service in Ilbe discussions were transvaluated to a morally enhanced sacrifice, consistent with *ressentiment*. Through neoliberalist personal responsibility, under a meritocratic system demanding constant self-improvement, these young South Korean men appear to have felt victimised by the conscription system. Consistent with Agostino (1998), their masculinity appeared to be shaped through 'physical discipline where male bodies are viewed [. . . ] as superior to female bodies' (p. 58), and through that superiority, men go through a 'metamorphosis into soldiers [that] involves their separation from 'effeminate civilians'' (Moon 2005, p. 72). In the South Korean context, conscription on Ilbe was mentioned as a key component to constructing male identities through the isolation of the masculine. At the same time, conscription instilled aggres-

sion and banished femininity, marking women and by extension feminism as an identifiable re-attachment for feelings of inferiority, injustice, and victimisation.

The banishment of the female form and effeminate character in an isolated gendered setting on the Ilbe forum creates women as both 'other' and 'inferior', making them an easily identifiable external object when re-attaching feelings of impotence or inferiority. The product of this was aggression towards feminists and women not only in the Ilbe data, but also in the backlash to the Mokdong/Jinmyung girls school incident, where some young men advocated for violence and sexual assault against the young women who wrote derogatory letters to conscripts (see Appendix C). We see again evidence that militarised masculinity breeds aggression in a gendered setting (Choi 1997), making women and feminists the most identifiable 'other' to direct negative feelings towards.

In addition to women, frustrations were directed towards the political elite. When discussing discontent for conscription among young men in South Korea, Moon describes a generation that is 'individualistic and unwilling to sacrifice' (Moon 2005, p. 70). This is considered a product of the hyper-competitive meritocratic model within a neoliberal society that South Korea has evolved into, where young South Koreans are unable to accept enforced collectivism at the hindrance of their self-development (Choi and Kim 2016; Kim and Finch 2022). In our data, too, young South Korean men on Ilbe used military conscription references to express feeling disadvantaged compared to women, pointing to increased feelings of victimhood. By not being able to see the barriers that marginalised groups face in South Korea, these young men, from a position of privilege, appeared to have seen only their group as the victims of the conscription system, and some placed the blame on the political elite.

Victimhood was often expressed on Ilbe in conversations about sex. The literature on feelings of victimhood among men due to sexual impotence, whether discussing a change in sexual capital or market value (Illouz 2019; Bloodworth 2020) or narcissistic regressions of white supremacy (Hoffman et al. 2020), are rooted in a perceived pre-existing entitlement being eroded or taken away (Bratich and Banet-Weiser 2019). Like Ilbe, other Incel networks are 'dedicated not only to the accumulating of more erotic capital for men, but also recouping that capital that has been lost' (Bratich and Banet-Weiser 2019, p. 5008). By sharing and discussing explicit content, these networks confirm a male entitlement of women presenting attractively to them, while highlighting that feminists are denying this entitlement—leading to feelings of victimhood.

Bloodworth (2020) and Illouz (2019) note that negative emotions originate from men perceiving themselves as having lower sexual capital or market value. We found evidence of this dynamic in the Ilbe data: young South Korean men perceived themselves as having a lower sexual market value, devalued themselves, felt inferior, and ultimately projected negative affects to women–enemies, retaining for themselves a righteous victimhood. As such, the Incel culture in South Korea is not just about gender, but appears to contain considerations of race and nationality, expressed as sexual impotence and inferiority. These complexities of sexual capital and market value appear in our data to have led young men to feeling victimised through the denial of desire for physical intimacy or sexual appeal. The women-enemies were presented (and probably perceived) as denying these young men their assumed original entitlement of having women as sexual objects.

For our empirical plausibility probe, we relied on a small number of survey participants and the content analysis of sampled material from far-right Incel blog Ilbe. While our analysis provides useful insights about the psychological undercurrent of the expressed frustrations and grievances of young South Korean men and offers the quantitative and qualitative measures as a useful toolkit for capturing *ressentiment,* they are by no means indicative of population trends or generalizable to all young men in South Korea. Further data collection of large and representative survey samples and expansion of the coding analysis to more cases and a wider timeframe can allow more confident generalisation of the findings.

We can reflect, however, on the key findings: in the grip of *ressentiment*, the young men discussing their frustrations in Ilbe changed the value of military service from a hardship into a morally righteous sacrifice. As such, the narratives of success and failure we examined were weak in agency or desire to propose a counter-narrative of resistance to military conscription. The powerlessness and passivity of *ressentiment* can be interrogated further with interviews which would allow delving deeper in the contents of their narratives, and examine whether these narratives are self-recognised, thus internalised as individual or collective identities. Our findings about conscription also warrant further investigation. Grievances about conscription were mentioned frequently on Ilbe but not in the qualitative survey. We think the online Ilbe environment facilitated sharing narratives of victimhood about conscription and redirected negative affects onto 'enemy' others. We also think the strictures of the COVID-19 pandemic heightened the expressed sense of victimhood about military conscription, contributing to emotionally loaded, heavily politicised, gendered and conspiracy-laden narratives. The effects of the pandemic have been studied in relation to mental health and depression, and studies can further explore their link with grievances and frustrations against other groups in society.

While our analysis was limited to survey and Ilbe data, our coding framework can be applied to films, songs, fiction, and comics. It would be interesting to examine in parallel the dominant political narratives of the party and leaders in power in South Korea, either via documents or interviews, and assess whether they are alleviating, reinforcing, or capitalising on these grievances for the next electoral contest. Many far-right organisations operate in similar online ecosystems to Incel groups; with this in mind, our analysis can expand to such networks and study their accounts of success and failure during the pandemic, identifying affective and cognitive 'meeting points' with the Ilbe users.

## 5. Conclusions

In this article, we examined anti-feminist, anti-women, anti-globalisation, and anti-military conscription accounts among a small number of survey responses of young South Korean men and users of Ilbe, applying the theoretical framework of *ressentiment* to identify patterns in our data. Powerlessness, victimhood, and indignation were evident in accounts of individuals striving for self-improvement and material gain, discussions of military conscription, and sex. Putting these findings in context, extant studies note that in South Korea neoliberalism stimulates a hyper-competitive meritocracy. We found these conditions intensify *ressentiment* affects. Scholars document a shift in the last decade among youth, where success and aspiration focus on personal and material gain, rather than personal attainment for a collective goal (Abelmann et al. 2009). Entrenched meritocratic ideals mean upwards mobility is seen as attainable if one only works hard enough. This personal responsibility of self-development comes hand in hand with a lack of contentment. There will always be a measurable or ambiguous goal ahead, meaning that there is always something to be denied, sparking envy, shame, and inefficacious anger. When coupled with the inability to recognise societal inequalities beyond oneself, these 'incomplete successes' trigger individual perceptions of injustice and failure. When there is no clear scapegoat to be blamed for negative affects, feelings of individual and collective moral victimhood gain ground. This misidentified victimhood mentality among the sample of young South Korean men we examined is sustained through an online network of misogyny—ultimately making it increasingly difficult to diffuse.

The research we undertook generated material on identity, agency, and subjectivity. We looked at the young men we studied and their position in South Korean society dialectically, examining how their socio-political environment determined their experiences and how they, in turn, responded to this environment. We sought to identify whether the experiences of the few young men we surveyed were linked to wider narratives of *ressentiment*, recognising that they do not 'speak on behalf of' all young South Korean men, and that their reflections were made at a specific point in time in their lives and are therefore partial and subject to change. We are also conscious of the importance of meaning, inter-

pretation, and power in our research, and the value of exercising reflexivity. We recognise geographical variations in masculinity and femininity, the social construction of gender, success, and failure. As we conducted this research, we took care to allow the voices of young men (through their statements) to tell their story, rather than us interpreting their words. Having their written responses allowed us to go back and check to make sure we have not imposed meaning over their words. Gender affected the way we conducted our research (opting for surveys to allow for anonymity and less intrusion), and we took care how we worded the questions in our questionnaire to be respectful and unintrusive. Recognising that language is power, we invited our participants to complete the questionnaire in the language they preferred (Korean or English), and we checked translations with a native Korean speaker to avoid misinterpretations of culturally loaded terms. Finally, we reflected on the insider/outsider dynamic in our study, and we took care to read widely on South Korea's cultural, social, economic, cultural, and political context to deepen our understanding. Central to our research has been the ethical obligation to not (re)produce prejudicial representations and instead allow deeper and fairer appreciation of the phenomena we studied.

This research invites us to think about the social and cultural specificities that can generate narratives of success and failure in *ressentiment*. The personal responsibility of success founded on consumption ability ('enjoy small shopping and entertainment') was the marker for success among those individuals *in ressentiment*, whereas the personal responsibility to provide for others in a non-consumer-centric sense was more prominent among those with lower *ressentiment*. Understanding anti-narratives, stories of success and failure, frustrations, and grievances is as much a psychological as it is a cultural and a political affair. Here, we looked at their intersection, focusing on the psychology of what is coveted, desired, wished, and denied, by studying individuals' accounts of success and failure in a context of neoliberalism and heightened meritocracy. What we found is that *ressentiment*, like other psychological processes, is also socially constructed and shared and rooted in other social artefacts (Berger and Luckmann 1966). Our study took place during the COVID-19 pandemic, and we found traces of its social impacts in how young South Korean people internalised feminism-related content and produced emotionally loaded and gendered narratives.

As 'narrative and imagination are integrally tied to one another' (Andrews 2014, p. 1), our findings generate new questions shared with other contributions: what type of polity do the holders of *ressentiment* accounts of success and failure imagine (see McAuley and Nesbitt-Larking 2022), and how do their ideals relate to the political offerings by parties and their leaders who seek to harness support in contemporary political competitions? For example, gendered nationalist narratives involve notions of the ideal woman as a mother and a nurturer of the motherland/nation (see Kinnvall and Singh 2022). Such dominant narratives promoting manhood and strength do not align with the powerlessness of the *ressentiment* narratives we identified here, which discredit womanhood while victimising men. We can then ask further, if what is politically on offer accentuates *ressentiment* among young men, would they be able to engage in dialogue with other narratives, and could their growing disaffections impart political impact that is socially responsible (see Phoenix 2022)? For example, could the contentious issue of mandatory conscription become the focus of public debate or lead to the development of counter-narratives that can resist the status quo, instead of transvaluated frustration internalised as morally enhanced sacrifice?

Because *ressentiment* is property of the powerless and the marginalised, and since marginalisation and exclusion breed radicalisation (Pretus et al. 2018; Kisić Merino et al. 2020), the prospects of *ressentiment* yielding pro-social outcomes are slim. A different emotional mechanism would be essential for socially responsible scenarios to take effect. We noted the intensification of misogynistic appeals in online communities like Ilbe where young South Korean men feel they are not alone. While these platforms often serve as digital echo chambers, reinforcing grievances, promoting radicalisation, and inciting violence, they also offer an opportunity to vent frustrations. If venting transforms into shar-

ing among like-minded peers, these narratives could become descending voices with the power to mobilise political action. Even among the most impoverished, hope for change is possible, when its passion is shared in a like-minded collective, as research on social movements astutely demonstrates (Salmela and von Scheve 2018).

We paraphrase Bradbury (2020) as we recognise that personal, social, psychological, and political experiences, imaginations, and their narratives are intertwined and constantly in flux. This is a valuable insight for scholars and policy practitioners because it deters deterministic accounts containing demonisation, stigmatisation, and dehumanisation, which can harden the victimhood mentality of groups and result in further exclusion. Our analysis shows that investing in the systematic study of *ressentiment*, identified in narratives and embedded in cultural and political contexts, can shed light on the possibilities for change and resistance in realities that often seem pre-destined for destructive outcomes.

**Author Contributions:** Conceptualization, T.C., E.N. and M.S.; methodology, T.C. and E.N.; software, T.C. and E.N.; validation, T.C., E.N. and M.S.; formal analysis, T.C. and E.N.; investigation, T.C. and E.N.; resources, T.C., E.N. and M.S.; data curation, E.N.; writing—original draft preparation, T.C., E.N. and M.S.; writing—review and editing, T.C., E.N. and M.S.; visualization, T.C., E.N and M.S.; supervision, T.C.; project administration, E.N.; funding acquisition, none. All authors have read and agreed to the published version of the manuscript.

**Funding:** Salmela's work was supported by funding from the European Research Council (ERC) under the European Union's Horizon 2020 research and innovation program (grant agreement No. 832940). No other funding was received for the research.

**Institutional Review Board Statement:** The study was conducted in accordance with the Declaration of Helsinki, and received Ethics approval by the University of Birmingham.

**Informed Consent Statement:** Informed consent was obtained from all subjects involved in the study.

**Data Availability Statement:** The data is available by the authors upon request.

**Conflicts of Interest:** The authors declare no conflict of interest.

## Appendix A. Questionnaire and Survey Questions

*Research into South Korean Society*

Thank you so much for agreeing to be surveyed as part of this research project. Please try to write answers in English but you may respond in Korean if that is more comfortable. All of your responses are strictly confidential, and your name will not be linked to any of the research materials. You will not be identified or identifiable in the report or reports that result from this research. With all of this in mind, do you agree to continue?

이 연구 프로젝트의 설문조사에 응해주셔서 감사합니다. 답변은 영어로 작성해 주시고, 한국어가 더 편한 경우에는 한국어로 답변해 주시기 바랍니다. 귀하의 모든 응답은 엄격히 비밀이 보장되며 귀하의 개인정보는 어떠한 연구 자료에도 연결되지 않습니다. 이 연구의 보고서나 파생되는 보고서에서 귀하를 식별할 수 없습니다. 이 모든 것을 염두에 두고 계속하는 데 동의하십니까?

- Yes, I agree—동의합니다
- No, I do not agree—동의하지 않습니다

Age

나이

- 18–20
- 21–24
- 25–30
- 31–35
- 36–40
- 41+
- Prefer not to say—말하지 않는 것을 선호

Gender
성별

- Male—남자
- Female—여자
- Prefer not to say—말하지 않는 것을 선호
- Other—다른

The next section is a little tricky, the questions are vague and open-ended. Before you start, take a second to think about yourself, who you are, who you want to be and how society is around you. Try not to think too hard about these questions and go with your gut instinct when you read them, they are not trying to trick you!
다음 섹션은 약간 까다롭고 질문이 모호하고 개방형입니다. 시작하기 전에 잠시 시간을 내어 자신, 자신, 되고 싶은 사람, 사회가 주변 환경에 대해 생각해 보십시오. 이 질문에 대해 너무 어렵게 생각하지 말고 읽을 때 직감대로 가십시오. 질문은 당신을 속이려는 것이 아닙니다!

Please tell me a little about yourself.
간단히 자기소개를 해주세요.

What are some of the things you would say are unique to South Korea?
한국만의 특징이라고 한다면 어떤 것이 있습니까?

What is your role in society?
사회에서 당신의 역할은 무엇입니까?
What are the roles of others in society?
사회에서 다른 사람들의 역할은 무엇입니까?

What is politics like in South Korea?
한국의 정치는 어떤가요?

What are the issues most important to you?
당신이 관심 있어하는 사회적인 문제/ 이슈는 무엇입니까?

What are your aspirations? Do they feel achievable?
당신의 포부는 무엇입니까? 달성 가능하다고 느끼는가?

How would you compare yourself against others you know?
자신을 타인과 비교할때 어떤 가치를 기준으로 비교합니까?

How would you define success?
성공을 어떻게 정의하시겠습니까?

How would you define failure?
실패를 어떻게 정의하시겠습니까?

How would you define fairness?
공정성을 어떻게 정의하시겠습니까?

What have been some of your recent challenges and what steps have you taken to overcome them?
최근의 어려움은 무엇이고 그것을 극복하기 위해 어떤 조치를 취했습니까?

This section will give statements to which you can rate how much you agree with the statement on a scale from 1 to 10. 1 = do not agree at all and 10 = fully agree.
이 섹션은 귀하가 진술에 동의하는 정도를 1 에서 10 까지의 척도로 평가할 수 있는 진술을 제공합니다. 1 = 전혀 동의하지 않음 및 10 = 전적으로 동의합니다.

Other people enjoy a better standard of living with less effort.
다른 사람들은 적은 노력으로 더 나은 생활 수준을 누립니다.

I feel that many people take advantage of my kindness.
많은 사람들이 나의 친절함을 이용하는 것 같아.

I feel I do not have people's respect.
나는 사람들의 존경을 받지 못한다고 느낀다.

Many seem important but they should not get such attention.
많은 사람들이 중요해 보이지만 그러한 관심을 받아서는 안 됩니다.

Everything goes wrong in my life, why me?
내 인생에서 모든 것이 잘못됩니다. 왜 나올까요?

My hopes and dreams will never come true.
내 희망과 꿈은 절대 이루어지지 않을거야.

Thank you for your responses. Do you agree to be contacted to discuss more about South Korean society?
응답해주셔서 감사합니다. 한국 사회에 대해 더 많이 논의하기 위해 연락하는 데 동의하십니까?

- Yes—네
- No—아니요
- Maybe—아마도

**Appendix B. Content Analysis Coding Variables**

1. Case No.—Case number
2. Ingroup mentioned—Is there an in-group mentioned/inferred?
3. Outgroup mentioned—Is there an outgroup mentioned/inferred?
4. Nostalgic thinking—Are there traces of nostalgic thinking (If yes, copy text)
5. Desire for change—Is there a desire for change? (If yes, copy text)
6. Change Forwards—Is there a desire for change forwards/in the future? (If yes, copy text)
7. Change Backward—Is there a desire for change backwards/towards the past? (If yes, copy text)
8. Abrupt change—Is there a desire for abrupt change? (If yes, copy text)
9. Anti-positions—Are there any anti-positions? (If yes, copy text)
10. Hope—Are there traces of hope? (If yes, copy text)
11. Efficacy—Are there traces of efficacy? (If yes, copy text)
12. Indignation—Are there traces of indignation? (If yes, copy text)
13. Transvaluation—Are there traces of transvaluation? (If yes, copy text)
14. Victimhood—Are there traces of victimhood? (If yes, copy text)
15. Powerlessness—Are there traces of powerlessness? (If yes, copy text)
16. Injustice—Are there traces of injustice? (If yes, copy text)
17. Destiny—Are there traces of destiny (If yes, copy text)
18. Violent/Illegal Action—Is there reference to violent or illegal actions? (If yes, copy text)
19. OwnGroupBest—Is one's own group described as the best? (If yes, copy text)
20. GroupUnderappreciated—Is one's own group described as underappreciated? (If yes, copy text)
21. Topic markers—What are the topics/themes discussed?
22. Total—The total presence of all markers
23. Total *ressentiment* markers—The total of the core *ressentiment* markers (coding variables 12–17)

## Appendix C. Content Analysis Coding Examples

**Table A1.** Extracts.

| Case No. | Image | Translation | Text | Translation |
|---|---|---|---|---|
| 1 |  | N/A | 옆에 붙어 있던 게 문재앙이었음 | Moon Jae-ang was next to me |
| | | | 역사적으로 의미도 없는 여자 사진 한장 | A historically meaningless picture of a woman |
| | | | 세종대왕보다 위대한 5만원 신사임당 | Shin Saimdang of 50,000 won greater than King Sejong the Great |
| 2 | N/A | N/A | 예전에 이런 일화가 많잖아? 아들, 딸이 같이 있는 집안은 거의 아들만 공부 시키고 딸이 무슨 공부냐? 이러면서 학교도 못 나온 딸들 많았는데 지금 돌아가는 거 보면 오히려 이게 ㄹㅇ 혜안이였음. | Have you had a lot of stories like this in the past? In a family with a son and a daughter, almost only the son studies, and what kind of study does the daughter do? There were a lot of daughters who couldn't even go to school while doing this, but looking back now, this was a real insight. |
| | | | 여자가 학력이 올라갈수록 결혼, 출산율은 반대로 운지함. 그리고 학력높은 여자일수록 ㅅㅂ 꼴페미 확률이 오히려 올라감. | Marriage and fertility rates reversed as a woman's education level increased. And the higher the educational level, the higher the probability of being a [disgusting] radical feminist. |
| | | | 남자 명문대 여자지잡 | Men's Universities, Women's Ji-job [non-prestigious university graduate] |
| | | | 남자 지잡 여자 고졸 | Male Ji-Job [non-prestigious university graduate], Female High School Graduate |
| | | | 남자 고졸 여자 중졸 | Male high school graduate, Female middle school graduate |
| | | | 이런 식으로 남자는 본인보다 한단계 아니 두단계 낮은 년도 마음먹고 외모 괜찮으면 결혼하는 남자들 꽤 되는데 보지년들은 절대 아님. | In this way, there are quite a few men who get married if they think they're one or two steps lower than themselves, and if they look good, they're definitely not pussy bitches. |

**Table A1.** *Cont.*

| Case No. | Image | Translation | Text | Translation |
|---|---|---|---|---|
| 2 | N/A | N/A | 니들 여자 대졸 남자 고졸 커플 본적있냐? | Have you ever seen a female college graduate male high school graduate couple? |
| | | | 출산률 올리려면 옛 어르신들 선견지명 대로 여자는 초등학교까지만 배우게 하고 학교 못 다니게 해야된다. 여자가 더하기 빼기만 할 줄 알면되지 뭘 더 배우려고 하노? | In order to raise the fertility rate, as the old elders had foreseen, women should only learn up to elementary school and be barred from attending school. A woman only needs to know how to add and subtract. What more are you trying to learn? |
| 3 |  | N/A | 자본주의 맛보고 살 뒤지게 쪄서 인생 나락가니 | Life goes downhill after tasting capitalism and gaining too much weight |
| | | | 페미니즘 외치며 국민들 경제력 이것저것 뺏어가려고 개지랄떠네 ㅋㅋ | They shout feminism and try to steal the people's economic power. |

**Table A2.** Coding examples of 3 cases.

| Coding Variable | Case 1 | Case 2 | Case 3 |
|---|---|---|---|
| Ingroup mentioned | 0 | Men | The public/people |
| Outgroup mentioned | 0 | Women | Capitalists/feminists |
| Nostalgic thinking | A historically meaningless picture of a woman Shin Saimdang of 50,000 won greater than King Sejong the Great | In order to raise the fertility rate, as the old elders had foreseen, women should only learn up to elementary school and be barred from attending school | 0 |
| Desire for change | 0 | In order to raise the fertility rate, as the old elders had foreseen, women should only learn up to elementary school and be barred from attending school | 0 |
| Change forwards | 0 | 0 | 0 |
| Change backwards | 0 | In order to raise the fertility rate, as the old elders had foreseen, women should only learn up to elementary school and be barred from attending school | 0 |

**Table A2.** *Cont.*

| Coding Variable | Case 1 | Case 2 | Case 3 |
|---|---|---|---|
| Abrupt change | 0 | 0 | 0 |
| Anti-positions | 0 | Anti-feminist | Anti-capitalist, anti-feminist |
| Hope | 0 | 0 | 0 |
| Efficacy | 0 | 0 | 0 |
| Envy/Jealousy | A historically meaningless picture of a woman Shin Saimdang of 50,000 won greater than King Sejong the Great | There are quite a few men who get married if they think they're one or two steps lower than themselves, and if they look good, they're definitely not pussy bitches. | 0 |
| Transvaluation | A historically meaningless picture of a woman | 0 | 0 |
| Victimisation | 0 | 0 | They shout feminism and try to steal the people's economic power. |
| Powerlessness | 0 | 0 | Life goes downhill |
| Injustice | 0 | 0 | Steal the people's economic power |
| Destiny/Action | 0 | As the old elders had foreseen | 0 |
| Violent/Illegal | 0 | 0 | 0 |
| OwnGroupBest | 0 | 0 | 0 |
| OwnGroupUnderappreciated | Shin Saimdang of 50,000 won greater than King Sejong the Great | 0 | 0 |
| Topic markers | Economy, Chosun (Sejong) | Education, marriage | Capitalism, economy |
| Total | 5 | 8 | 7 |
| Total *ressentiment* markers (12–17) | 2 | 2 | 3 |

## Appendix D. Tables

**Table A3.** Content Analysis 'Sex' data.

| | | Cases Referring to Sex, Sexual Appeal, or Rape (32%) | Sample Overall |
|---|---|---|---|
| Reference to race or nationality | | 50% | 42% |
| Blame or outgroup mentioned | Feminists | 69% | 70% |
| | Women | 25% | 14% |
| | Feminists and women combined | 94% | 82% |
| Indicators of *ressentiment* present (average) | | 38% | 29% |

**Table A4.** Sources of Victimhood and Discontent in Conscription Cases.

| Case No. | Perceived Injustice | Reattachment Defence, What/Who Is to Blame? |
|---|---|---|
| 4 | 'I almost think that I am the lowest class in society [. . . ] I wake up every day at minus 20 degrees, 40 degrees below a sensible temperature and I freeze up and work for several hours at my post [. . . ] No matter how much service time, I still write a letter. If the social atmosphere had the least respect' | 'It was Moon Jae-in who gave dog food to soldiers quarantined due to Corona<br><br>Moon Jae-in is a bastard' |
| 17 | 'The incident of condolence letters at Jinmyung Girls' High School' | 'The feminists' banners hahahaha. . .<br>What a mess. . . ' |
| 22 | 'Although Seok-Yeol Yoon is level 1 (economic and security right wing, liberal democratic centre-right, socioculturally centre-left), his opponent is so populist that he himself often shows "populist tendencies pushed back". (Soldier salary 2M [2 million won, approx. £1260])' | 'his opponent [Lee Jay-Myeong—Democratic Party] is so populist that he himself often shows "populist tendencies pushed back".' |
| 23 | '2 million won for a soldier. [2 million won monthly military salary, approx. £1260]' | 'Ahn Cheol-soo, this asshole' |
| 24 | 'Women mocking soldiers and blaming the men for their hardships (ex. the men made the army~)' | 'women' |
| 27 | 'Mokdong Academy' [condolence letters incident] | 'Femmy, who still hasn't cured her mental illness' |
| 29 | 'Why women have no choice but to demean the military. . . Feminists, above all else, demean, ridicule, and caricature men's military duty.<br>There is a need to nullify the sacrifices and hard work of the soldiers.' | 'Feminists [. . . ] women' |
| 46 | 'Soldiers won't be subjected to poor meals or ridicule. [If adopted as the 51st state of the USA] They spend a day working hard on the white horses they like and they can buy and eat in Pakchon with the dollars they receive.' | 'If you look at it [South Korea] now, both the Democratic Party and the National Power Party seem to have no answer' |
| 50 | 'Conscription itself is communism.' | 'The Feminist movement [. . . and] the government is the enemy of the people' |

## Notes

[1] Respondents were asked: 'Please tell me a little about yourself'; 'What is your role in society'; 'What are the roles of others in society'; 'What are some of the things you would say are unique to South Korea'; 'What is politics like in South Korea'.

[2] 'How would you define success'; 'How would you define failure'; 'How would you define fairness'; 'What are the issues most important to you'; 'What are your aspirations? Do they feel achievable'; 'What have been some of your recent challenges and what steps have you taken to overcome them'; 'How would you compare yourself to others you know'.

[3] Some posts were deleted from the site before the content could be viewed. It is possible that these posts included important data that could change the results and should be taken into consideration.

[4] Not a verified source but comprehensive information on the incident and backlash can be found here: https://www.koreaboo.com/news/girls-high-school-feminists-male-extremists-soldiers-letters/ (accessed on 2 January 2023).

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
