# Peer review of "Narratives of Success and Failure in Ressentiment: Assuming Victimhood and Transmuting Frustration among Young Korean Men"

_socsci, doi:10.3390/socsci12050259_

Round 1

Reviewer 1 Report

The approach using the concept and analysis tool of "Ressentiment" is this articles’ originality. However, the following points should be considered again.

1.

The phrase "during the Covid-19 Pandemic" in the title doesn't seem appropriate. This article did not interpret the results in consideration of the 'context' of the global pandemic. Social isolation due to COVID could negatively impact young men's lives or increase digital media uses, but the impact of the pandemic on men's digital media consumption patterns or changes in the way ressentiment is structured has not been explained in South Korea. Above all, the pandemic situation does not seem to be the main concern of this article.

2.

The purpose of this study is to investigate how the complex emotion of ressentiment is constructed. And the mechanism is mainly explained through the contents of Ilbe. If so, the difference from previous studies that analyzed the narratives of young men's narratives of victimhood or weakness and the findings of this article should be more clearly presented.

3.

It is also emphasized that this article will explain the mechanism of ressentiment through the 'narrative of success and failure'. Nevertheless, the results deal just briefly about the perceptions of success among qualitative survey respondents. The analysis of Ilbe was not linked to explanations of 'success and failure' in it(e.g., ressentiment for conscription system means failure for what? Or how is the ressentiment around conscription captured in the narratives of the success or failure of what?). It seems necessary to explain what success or failure is in order to claim that ressentiment is captured through the narratives of ‘success and failure’. 

3.

Considering the above, I am not sure whether an appropriate method for the purpose of the study has been used. Responses from qualitative surveys are treated as fragmented and are rarely used to explain the ‘relationship’ or ‘mechanism’ between each element. It is necessary to explain why the qualitative survey was conducted. It seems that it was a means to establish the prove for the narratives of ‘success and failure’ emphasized in the title, but the response content of the qualitative survey is not actually used to explain the constitution of mechanism of ressentiment.

Combining certain responses from another respondents’(the qualitative survey’s respondent and Ilbe's) is also problematic (for example, speaking of marriage and financial stability as success as in the qualitative survey does not necessarily lead to frustration as an Ilbe’s narratives. Even most of women or another generatioin would have responded similarly to the qualitative survey). In such an approach, young Korean men are viewed as a homogeneous group that is affected in the same way by structural pressures.

4.

“Anti-globalization” and “toxic-masculinities” are abstract concepts that can be explained in various aspects, so it seems that a clearer expression is needed(e.g., how is it different “anti-globalization” from the word “nationalistic” which is used in result?)

5.

“R” and “C” are used as symbols to separate cases, but I think that some of them are written incorrectly (please check last paragraph on page 8).

Author Response

Thank you for your very helpful comments. We have addressed them in the attached document. We hope you will find our response and the changes to the manuscript satisfactory. 

Reviewer 2 Report

This is a pathbreaking and critically important extension of political narrative analysis into a country and a culture that has complex connections to the West. In general, non-North American and European settings are understudied and so the current contribution is most valuable.

The extension of the political psychology of ressentiment to the South Korean setting combines the broader context of neoliberalism, the rise of right-wing populism, ambivalence toward militarism, and toxic masculinity in the exigency of the Covid-19 pandemic. The results are generally positive, but the one omission from the existing manuscript is an elaboration of the impact of Covid-19 itself – if any – on the social relations under analysis. This does not require much further elaboration, but there should at least be a short passage in the Discussion section of the manuscript. Among other possibilities for content here are a summary of the restrictions and conditions set in South Korea regarding the pandemic and any comments or claims or references made by the young male interviewees or contributors to the social media site specifically about Covid-19 and its impact. If there were only a few or no such comments, then this too is notable and could be reported. In other words, what if anything does living in a time of Covid-19 do to the social forces and relations under analysis in the manuscript?

The theoretical and structural-contextual elements of the manuscript are well developed and supported with relevant literature. So too is the elaboration of the psychodynamics of ressentiment. There is a rich and valuable Discussion section on the relationships among militaristic socialization, anti-elitism, the cultivation of misogyny, and ressentiment (but see above on Covid-19).

Altogether, with the minor change I have suggested, this promises to be a valuable contribution and is close to being ready for publication.

Minor points:

1.   The conversion of currency from won in the first few lines of Section 3.1 needs to be consistent. One comparator currency should be chosen, and the Euro or American dollar are both possibilities.

2.   There is a typo in the second line above the start of Section 3.4.

3.   There is a hanging additional comma about halfway through the Conclusion.

4.   There is a misspelling of “Mahendran” toward the end of the fourth paragraph of the conclusion.

Author Response

(The authors gave the same response as above.)

Round 2

Reviewer 2 Report

Narratives of Success and Failure in Ressentiment: Assuming Victimhood and Transmuting Frustration among Young Korean Men

Second Review

The author(s) responded effectively and fully to the comments and suggestions made in my first review.

Specifically, there is now a clear and well contextualized explanation of the study in the context of the Covid-19 setting. 

This is further extended in a now fuller and richer discussion of the Covid-19 related responses to both the questionnaire and the Ilbe website. 

Final minor points:

Some final concluding comment on the Covid-19 relevance (no more than a sentence or two) would tie things up and should be added.

The entire manuscript contains a number of small errors of grammar, style, and typographical errors. It requires a thorough read and correction.

Author Response

Thank you for the opportunity to further revise our manuscript. This has given us the chance to correct a few minor errors and also recalibrate our introduction (without adding or removing content) to present more clearly the aims and objectives of the article. Please see attached file. 
